# Influence of aluminium distribution on the diffusion mechanisms and pairing of [Cu(NH₃)₂]⁺ complexes in Cu-CHA

Joachim D. Bjerregaard [1,2] ✉, Martin Votsmeier[2] & Henrik Grönbeck [1] ✉

The performance of Cu-exchanged chabazite (Cu-CHA) for the ammonia-assisted selective catalytic reduction of $NO_x$ (NH₃-SCR) depends critically on the presence of paired $[Cu(NH_3)_2]^+$ complexes. Here, a machine-learning force field augmented with long-range Coulomb interactions is developed to investigate the effect of Al-distribution and Cu-loading on the mobility and pairing of $[Cu(NH_3)_2]^+$ complexes. Performing unbiased and constrained molecular dynamics simulations, we obtain unique information inaccessible to first-principle calculations and experiments. The free energy barrier for $[Cu(NH_3)_2]^+$ diffusion between CHA-cages depends sensitively on both the local and distant Al-distribution. Importantly, certain Al-distributions and arrangements of neighboring $[Cu(NH_3)_2]^+$ and $NH_4^+$ cations make paired $[Cu(NH_3)_2]^+$ complexes exothermic with respect to separated configurations. Our results suggest that the NH₃-SCR activity can be enhanced by increasing the Cu-loading and Al-content. The dynamic interplay between $[Cu(NH_3)_2]^+$ and $NH_4^+$ diffusion is crucial for the $[Cu(NH_3)_2]^+$ mobility and stresses the need to explore large systems including long-range Coulomb interactions when studying diffusion of charged species in zeolites.

Heterogeneous catalysis is generally performed at elevated pressures and temperatures, which affect the structure and composition of the active material during the reaction conditions. Numerous examples have demonstrated that the dynamic response of catalyst materials to the reaction conditions is crucial for the catalyst performance[1–3]. One outstanding example of the dynamic character of a catalyst, is Cu-exchanged Chabazite (Cu-CHA), which is used for selective catalytic reduction of $NO_x$ using ammonia as reducing agent (NH₃-SCR). The overall reaction for NH₃-SCR of NO is

$$4\,NH_3 + 4\,NO + O_2 \rightarrow 4\,N_2 + 6\,H_2O \tag{1}$$

where NO and NH₃ couple to form N₂ and H₂O. The stoichiometry requires an O₂ molecule to accommodate the H atoms in NH₃. Cu-CHA has demonstrated high activity and selectivity in a wide temperature range[4,5] in combination with high hydrothermal stability[6,7]. Cu-CHA is a

small-pore zeolite (aluminosilicate) material with an Si/Al ratio commonly in the range of 5–30 and a Cu-loading of some weight percent.

Cu is in the absence of NH₃ bound to the zeolite framework forming, e.g., ZCuOH and Z₂Cu species, where Z represents the anionic Al-site in the framework. During low-temperature operation (<250 °C), the Cu ions are solvated by NH₃, forming $[Cu(NH_3)_2]^+$ complexes[8,9] that are detached from the framework. The detached complexes are floating in the CHA cages, while being tethered to the anionic Z-sites via long-ranged Coulomb interactions. The mobility of the floating $[Cu(NH_3)_2]^+$ complexes is crucial for low-temperature activity as two $[Cu(NH_3)_2]^+$ complexes are needed to activate $O_2$[10,11]. The subsequent coupling of NO and NH₃ occurs over a $[Cu_2(NH_3)_4O_2]^{2+}$ species. The need for two $[Cu(NH_3)_2]^+$ complexes for O₂ activation is indicated by a second-order dependence of the activity on the Cu-loading[12,13]. It should be noted that Cu-CHA can be synthesized from Cu₂O via

[1]Department of Physics and Competence Centre for Catalysis, Chalmers University of Technology, SE-412 96 Göteborg, Sweden. [2]Umicore AG & Co. KG, Rodenbacher Chaussee 4, 63457 Hanau, Germany. ✉e-mail: joabje@chalmers.se; ghj@chalmers.se

solid-state ion-exchange in an ammonia atmosphere[14], which is a process that is governed by the diffusion of $[Cu(NH_3)_2]^+$ and the concurrent counter diffusion of $NH_4^+$.

The mobility of Cu ions in Cu-CHA has experimentally been investigated using impedance spectroscopy[15]. The measurements reveal that an increased Cu-loading enhances the mobility of $Cu^+$ in the presence of $NH_3$, which is not the case in the absence of $NH_3$[15]. Moreover, electron paramagnetic resonance experiments indicate that the activity of the $NH_3$-SCR reaction is limited by $Cu^+$ diffusion at low Cu-loadings[16]. It was in ref. 16 suggested that the paring and separation of $[Cu(NH_3)_2]^+$ complexes occur with similar frequency as the catalytic turnover and that the $NH_3$-SCR-activity is partly limited by the pairing of Cu-complexes. It should be noted that the pairing of two $[Cu(NH_3)_2]^+$ complexes is supposed to be endothermic due to the repulsion between the $Cu^+$ cations[11,17,18].

The experimental evidence of the importance of $[Cu(NH_3)_2]^+$ diffusion for the SCR reaction has stimulated computational studies on the barriers for inter-cage diffusion of $[Cu(NH_3)_2]^+$ with focus on the effect of the local Al-distribution[11,14,18,19]. Density functional theory (DFT) simulations predict the diffusion of $[Cu(NH_3)_2]^+$ to a neighboring cage sharing an Al-ion, to have a free energy barrier of about 0.2 eV[18]. If $[Cu(NH_3)_2]^+$ diffuses to an adjacent cage without an Al-site, metadynamics[11] and umbrella sampling[18] estimate the free energy barriers to be 0.57 and 0.79 eV, respectively. The presence of additional molecules in the zeolite (NO and $NH_3$) have been predicted to increase the free energy barrier by ~0.3 eV[18].

Recently, operando X-ray adsorption was combined with statistical simulations, including complex diffusion, to study the pairing of Cu-complexes in Cu-CHA[19]. It was concluded that the pairing of Cu-complexes in samples with low Al-density primarily occurs within one CHA cage apart, suggesting that only Cu-complexes in two connecting CHA cages form pairs. However, for samples with a high Al density (Si/Al = 6), it was possible to form pairs also with complexes being separated by one CHA-cage[19].

DFT-based calculations and simulations are typically restricted to small unit cells and time scales in the range of pico-seconds. The emergence of machine-learning force fields (ML-FF) that parameterize the energy and forces is a promising route to enable simulations of large systems and extended time scales. The ML-FF approach was recently used by Millan et al.[20] to investigate the effect of framework composition and $NH_3$ concentration on the diffusion of $[Cu(NH_3)_2]^+$. $[Cu(NH_3)_2]^+$ was observed to move distances up to 30 Å in the presence of $NH_3$ during a 5 ns simulation. The long-range diffusion was connected to the

counter diffusion of protons in the form of $NH_4^+$. It was concluded that a high Al-density and high Cu-loading enhances the probability of pairing[20]. One issue with standard ML-FF is, however, the assumption of chemical locality. Bonds are assumed to be local, and interactions are only considered in a sphere with a specified cut-off radius. The assumption of locality is questionable for Cu-CHA as the cations in the system ($[Cu(NH_3)_2]^+$ and $NH_4^+$) interact with the Al-sites in the CHA framework via long-ranged Coulomb forces. Thus, the use of ML-FF to Cu-CHA should include iterative refinements to incorporate long-range interactions such as simple dispersion corrections and charges[21].

Given that the performance of Cu-CHA depends critically on the mobility and pairing of $[Cu(NH_3)_2]^+$, it becomes important to investigate the dependence of mobility on zeolite properties such as Si/Al ratio and Cu-loading. Moreover, it is important to explore to what extent the Al-distribution affects the stability of $[Cu(NH_3)_2]^+$ pairs. Information on mobility and stability is needed to connect catalyst performance with materials properties and provide guidelines for materials designed.

Herein, we are able to make clear connections between zeolite properties and the mobility and stability of $[Cu(NH_3)_2]^+$ pairs by developing an ML-FF augmented with long-range interactions. The inclusion of long-range interactions are made possible by the use of a recently implemented method, based on distributed charges[22]. Free energy barriers of $[Cu(NH_3)_2]^+$ are evaluated with metadynamics at experimentally relevant Si/Al ratios and Cu-loadings. In addition, unbiased simulations are performed to analyze the time evolution of the mean-square displacement, fraction, and lifetime of paired complexes as well as pair correlation functions. The simulations provide important information on how the local and distant Al-distribution affect the diffusion barriers, diffusion mechanisms, and stability of complex pairs.

## Results

The $[Cu(NH_3)_2]^+$ complexes interact with the framework ions and the anionic Al-sites via long-ranged Coulomb forces. Thus, it can be assumed that the dynamics and stability of $[Cu(NH_3)_2]^+$ complexes depend on the local and distant distribution of Al-ions in the CHA framework as well as the Si/Al and the Cu/Al ratio. We investigate how these materials' properties affect the function of the system by studying diffusion barriers and pair stability. Enhanced sampling techniques are used to understand how specific local and distant Al-distributions influence the uncorrelated and correlated mobility of $[Cu(NH_3)_2]^+$ and $NH_4^+$, whereas unbiased molecular dynamics of large systems provide information on averaged properties.

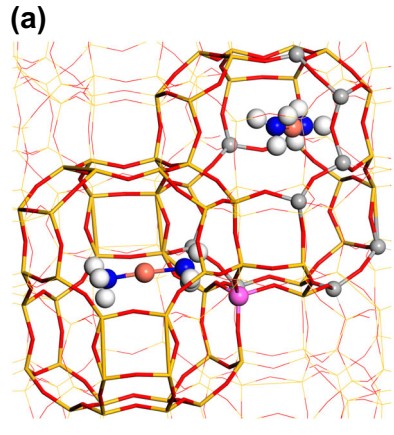

**(a)**

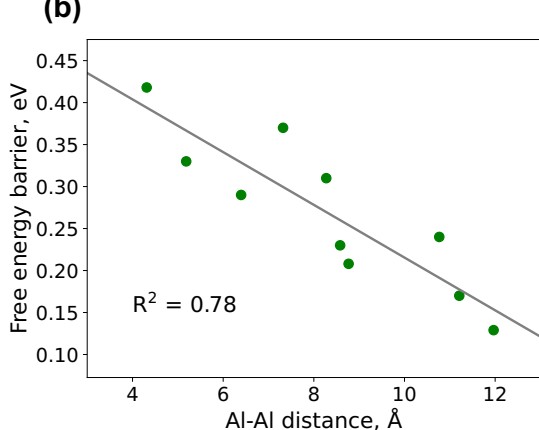

**(b)**

**Fig. 1 | Meta-dynamics simulation of the pairing of two $[Cu(NH_3)_2]^+$ complexes with different Al locations. a** Structure showing the Al-ion that is fixed (purple), and the different locations of Al (gray). Atomic color codes: H (white), N (blue), O (red), Al (purple and gray), Si (yellow), and Cu (bronze). $R^2$ is the coefficient of determination. **b** Correlation between Al-Al distance and free energy barriers. Source data are provided as a Source Data file.

## Effect of local and distant distribution of Al-ions

To investigate the effect of the local Al-distributions on the free energy barrier of the pairing of the $[Cu(NH_3)_2]^+$ complexes, we perform well-tempered metadynamics simulations using a $3 \times 2 \times 2$ supercell (one cell is the hexagonal unit cell with in total 36 Si and Al atoms) with periodic boundary conditions. To isolate the effect of the local environment, the system is considered with only two Al-ions. One Al-ion is placed in an eight-membered ring connecting two cages, whereas the second Al-ion is placed in different positions in the cage where the pair is formed. The location of the two Al-ions is shown in Fig. 1a. The free energy barrier for pairing is reported as a function of Al-Al distance, Fig. 1b. The free energy barrier is computed by taking the difference between the maximum and minimum points on the potential energy surface along the diffusion path. The free energy barrier ranges from 0.15 to 0.42 eV. The highest barriers are obtained for the short Al-Al distances, which is attributed to the repulsion between the two $[Cu(NH_3)_2]^+$ ions when the Al-ions are close. A high barrier is, furthermore, connected to a low stability of the formed pair and a low barrier for unpairing. The barrier for unpairing is calculated to be 0.02 and 0.13 eV for an Al-Al distance of 4.3 and 11.2 Å, respectively. The results show that the local distribution of Al-ions has a significant effect on the free energy barrier for $[Cu(NH_3)_2]^+$ diffusion.

Because of the long-ranged Coulomb interactions, the mobility and stability of $[Cu(NH_3)_2]^+$ depend not only on the local Al-distribution but also on the embedding environment, thus the distant distribution of Al-ions and the counter ions ($[Cu(NH_3)_2]^+$ and $NH_4^+$). The effect of the embedding environment is studied by well-tempered metadynamics simulations using a $3 \times 2 \times 2$ supercell. The simulation investigates the pairing of two $[Cu(NH_3)_2]^+$ complexes and the location of the Al-ions in the two cages in which the $[Cu(NH_3)_2]^+$ ion diffuses (local structure) is fixed, whereas the embedding environment is modified. The free energy landscapes are reported in Fig. 2a and the local structures are shown in 2b, c. A positive collective variable (CV) corresponds to paired $[Cu(NH_3)_2]^+$ complexes. The simulations are considered with an Si/Al ratio of 13 and 5, with the Cu/Al ratio set to 0.25 and different embedding environments are generated randomly. To prevent counter diffusion in the embedding environment, which would influence the free energy landscape, the motion of the $NH_4^+$ and $[Cu(NH_3)_2]^+$ ions in the embedding environment is restricted to a maximum distance of 5 and 6 Å, respectively, between the Al-ions and counter ion pair. The chosen limit is based on a pair correlation function (PCF) analysis (see Supplementary Fig. 9) and ensures that the counter ion can move freely with respect to the corresponding Al-ion.

For the simulations with an Si/Al ratio of 13, the estimated free energy barriers for two different embedding geometries with the local structure as in Fig. 2b are 0.44 (I) and 0.36 (II) eV and the paired states are 0.42 (I) and 0.29 (II) eV less stable. For an Si/Al ratio of 5, the free energy barriers are 0.31 (III) and 0.22 (IV) eV with the paired state being 0.21 (III) and 0.09 (IV) eV less stable. Hence, a lower Si/Al ratio in the embedding environment lowers the diffusion barrier and enhances the stability of paired $[Cu(NH_3)_2]^+$ complexes. The increased stability can be attributed to the Coulomb interaction resulting from the increased number of Al-ions in the embedding environment. Additional simulations are performed (marked *) for a Si/Al ratio of 13, with a local Al distribution not having an Al-ion in the connecting eight-membered ring, Fig. 2c. Hence, one of the $[Cu(NH_3)_2]^+$ complexes is far from the Al-ion in the paired state. The free energy barriers with this local configuration are significantly higher. Interestingly, the energy of the paired state in configuration V is 0.32 eV, which is stabilized with respect to configuration I. This shows that the embedding environment can significantly stabilize the paired $[Cu(NH_3)_2]^+$ complexes. The results stress the significance of the embedding environment and highlight that the pair stability depends not only on the local Al distribution, but also on the embedding environment.

## Correlated mobility of $[Cu(NH_3)_2]^+$ and $NH_4^+$

The cations ($[Cu(NH_3)_2]^+$ or $NH_4^+$) are tethered to the anionic Al-ions and long distance mobility requires exchange of cations to ensure local charge neutrality. This kind of cation exchange is referred to as counter diffusion and examples are shown in Fig. 3. Figure 3a, illustrates an example of counter diffusion-induced pairing, where the mobility of a $NH_4^+$ ion facilitates the pairing of $[Cu(NH_3)_2]^+$ complexes.

Two-dimensional metadynamics simulations are performed to explore the correlated mobility between two cations by introducing two collective variables. The embedding environment has an Si/Al of 13 and a Cu/Al ratio of 0.25. In similarity to the previous section, the cations in the embedding environment are constrained, allowing exploration of the correlation between the local mobility of $[Cu(NH_3)_2]^+$ and $NH_4^+$. The two-dimensional metadynamics simulation is shown in Fig. 4a, with selected lowest free energy paths in Fig. 4b. Six different minima (A–F) are obtained, and the schematic positions are shown in Fig. 4c. A positive value of CV for $[Cu(NH_3)_2]^+$ corresponds to paired $[Cu(NH_3)_2]^+$ complexes. If the CV for $NH_4^+$ is above 8, the free energy barrier for pairing (A → B) is 0.48 eV and the paired and separated states have similar stability. If instead, the $NH_4^+$ ion diffuses to the minima corresponding to a CV of about 4, the free

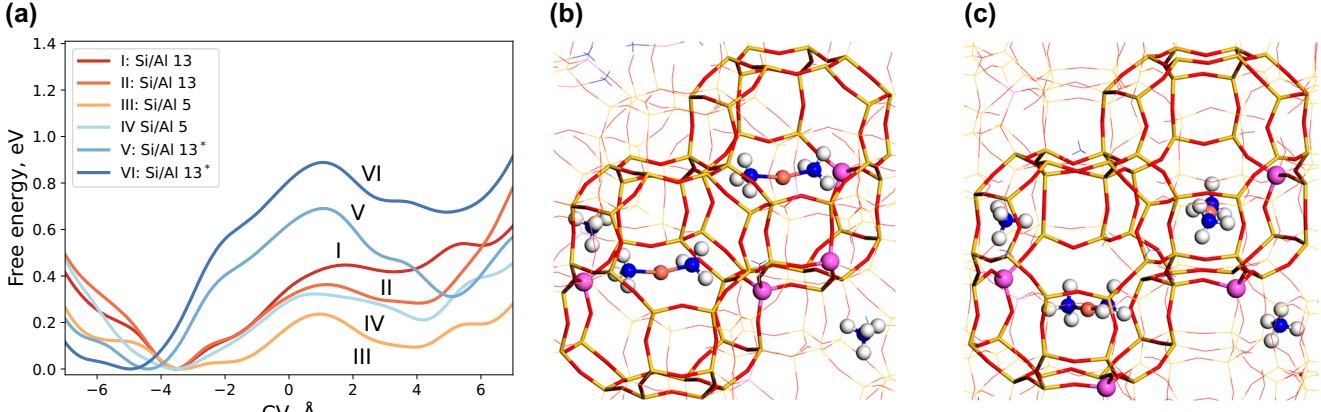

**Fig. 2 | Meta-dynamics simulations of the pairing of two $[Cu(NH_3)_2]^+$ complexes. a** Free energy profiles as a function of the collective variable (CV). **b** Structure with the local Al configuration with an Al-ion in the eight-membered ring. **c** Structure with the local Al configuration without an Al-ion in the eight-membered ring (marked with * in (**a**)). Atomic color codes: H (white), N (blue), O (red), Al (purple), Si (yellow), and Cu (bronze). Source data are provided as a Source Data file.

energy barrier is reduced to 0.42 eV and the paired state is preferred by −0.19 eV. The last energy profile (E → F) has a barrier of 0.71 eV and the separated state is preferred by 0.24 eV. The two-dimensional

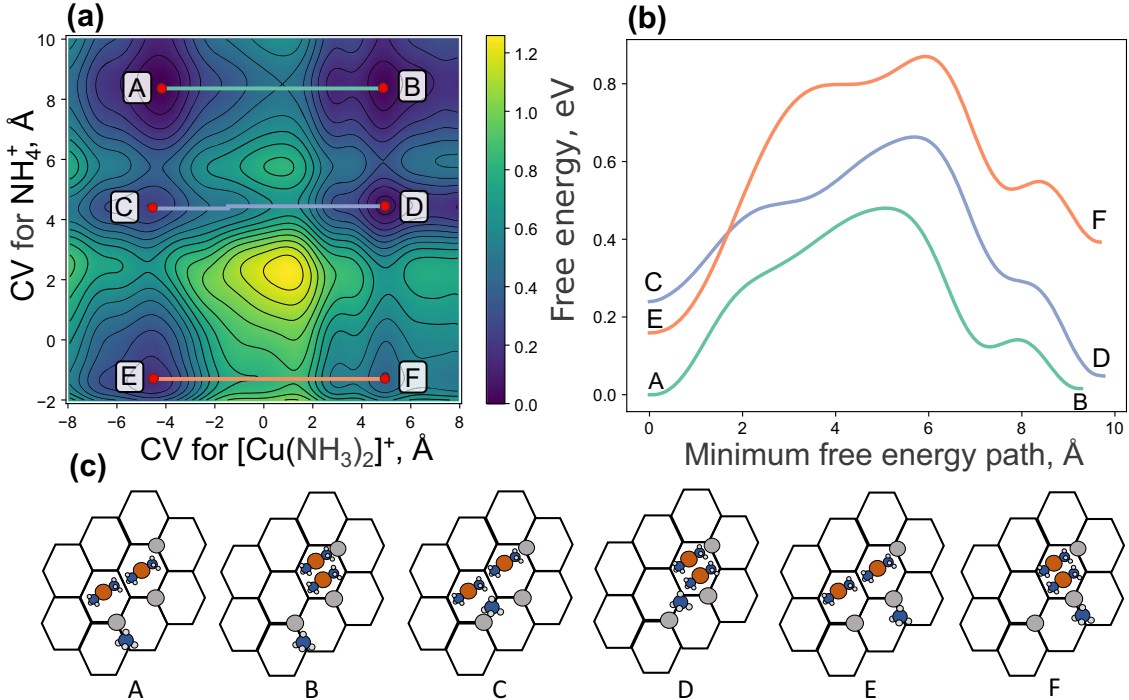

**Fig. 3 | Illustration of different diffusion processes. a** Counter diffusion-induced pairing, involving the diffusion of a $NH_4^+$ species in the neighboring cage. **b** Counter diffusion where a Cu-complex and $NH_4^+$ species switch location. **c** Counter diffusion, where the movement of a Cu-complex enables the further diffusion of a $NH_4^+$ ion.

metadynamics simulation clearly demonstrates that the position of the $NH_4^+$ ion has a crucial effect on both the barrier for the diffusion and the stability of the paired state. The free energy landscape underscores the correlated mobility of the cations in the system, which is a consequence of the long-ranged Coulomb interactions between anions and cations.

### Unbiased mobility and pairing of $[Cu(NH_3)_2]^+$

Having probed the effect of the local and embedding distribution of Al-ions on the free energy barriers, we perform unbiased MD simulations in a periodic $3 \times 3 \times 3$ supercell with two Si/Al ratios (5 and 13) and different Cu/Al ratios. Having an Si/Al ratio of 5 and Cu/Al ratios of 0.75, 0.5, and 0.25 corresponds to simulations with (121,41), (81,81), and (40,122) $[Cu(NH_3)_2]^+$ complexes and $NH_4^+$ species, respectively. The simulations are done to investigate the mobility and pairing dynamics of the $[Cu(NH_3)_2]^+$ ions. Figure 5a, b show the fraction of paired $[Cu(NH_3)_2]^+$ as a function of time (solid lines). A fraction of 1.0 corresponds to all $[Cu(NH_3)_2]^+$ complexes being paired. The dashed lines represent the average fraction of paired $[Cu(NH_3)_2]^+$ if the complexes were randomly distributed (see the Supplementary Methods for details on how the average fraction of paired Cu are computed).

In Fig. 5a, the pairing is shown for an Si/Al ratio of 5. A higher Cu/Al ratio leads to a higher fraction of paired $[Cu(NH_3)_2]^+$ complexes, in accordance with the estimates for randomly placed complexes. For Cu/Al ratios of 0.75 and 0.5, the simulated fractions of paired $[Cu(NH_3)_2]^+$ complexes are lower than if the complexes were randomly distributed. This suggests that the repulsion between the $[Cu(NH_3)_2]^+$ ions destabilize the paired state. For a Cu/Al ratio of 0.25, the simulated paired fraction is after ~2 ns instead higher than the randomly distributed case. This implies that a high fraction of $NH_4^+$ and Al-ions compared to $[Cu(NH_3)_2]^+$ can result in configurations with a high stability of the $[Cu(NH_3)_2]^+$ pair. A simulation with the same number of $[Cu(NH_3)_2]^+$ complexes but different amounts of $NH_4^+$ and

**Fig. 4 | Two-dimensional metadynamics simulation for the movement of $[Cu(NH_3)_2]^+$ and $NH_4^+$ ions with the embedding network having a Si/Al ratio of 13 and Cu/Al of 0.25. a** Free energy landscape as a function of the collective variables (CV), with relevant minima highlighted by a letter. **b** Selected free energy barriers for the minimum energy path between different minima, with the paths shown by colored lines in (**a**). **c** Placement of $[Cu(NH_3)_2]^+$ and $NH_4^+$ for the different minima (A–F) in (**a**). Source data are provided as a Source Data file.

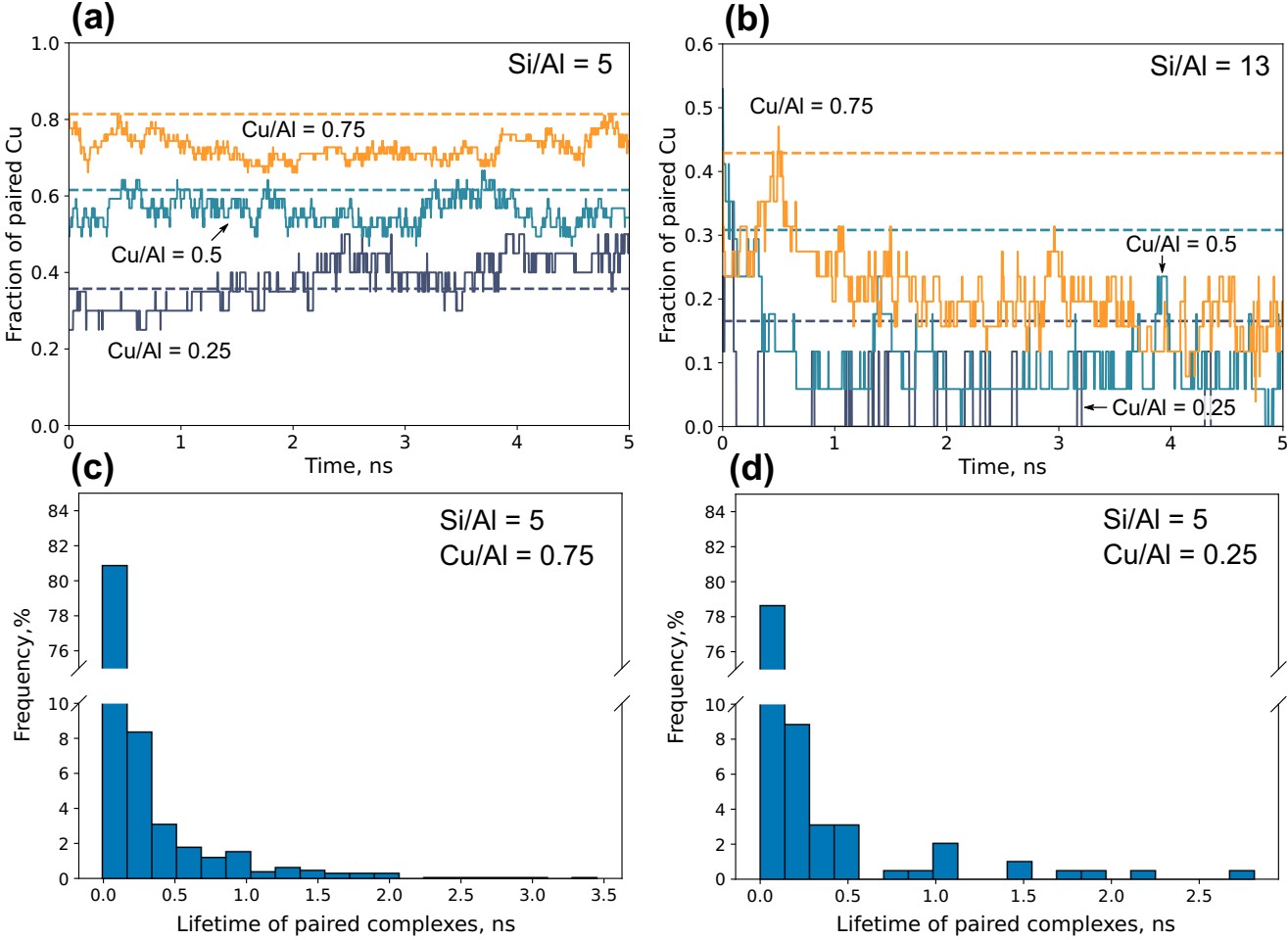

**Fig. 5 | Analysis of paired $[Cu(NH_3)_2]^+$ complexes from unbiased simulations.** Dashed lines represent the fraction of paired $[Cu(NH_3)_2]^+$ if the complexes were randomly distributed. The different lines represent Cu/Al ratios of 0.25, 0.5, and 0.75 with an Si/Al ratio of **a** 5 and **b** 13. Histogram computed for the lifetime of

paired $[Cu(NH_3)_2]^+$ complexes for Si/Al ratio of 5, with Cu/Al ratio of **c** 0.75 (1218 data points) and **d** 0.25 (192 data points). Source data are provided as a Source Data file.

Al-ions, confirm that a large fraction of Al-ions leads to a higher fraction of paired $[Cu(NH_3)_2]^+$ (see Supplementary Fig. 10).

The Si/Al ratio has a pronounced effect on the fraction of paired Cu-complexes, which is lowered as the Si/Al ratio is increased, see the case with an Si/Al ratio of 13 in Fig. 5b. The trend with respect to the Cu/Al ratio is the same as for an Si/Al of 5, namely that a higher Cu/Al ratio leads to a higher fraction of paired Cu. However, compared to a Si/Al ratio of 5, the simulated fraction of paired $[Cu(NH_3)_2]^+$ complexes is, in all cases, well below the estimate with randomly placed Cu-complexes (dashed lines). This shows the importance of collective effects where abundant Al-sites and $NH_4^+$ enhances the stability of paired complexes.

As $O_2$ activation requires a pair of $[Cu(NH_3)_2]^+$ complexes, it is interesting to investigate the lifetime of the pairs. The distribution of lifetimes are shown in Fig. 5c, d for Si/Al ratio of 5, with Cu/Al of 0.75 and 0.25, respectively. For Cu/Al = 0.75, most paired complexes have a lifetime of less than 0.1 ns, however, some have lifetimes of almost 3.5 ns. The large difference in lifetimes shows that some paired complexes adopt configurations with enhanced stability. The lifetime of the pairs can be related to the timescale of $O_2$ adsorption, which, according to collision theory, is about $10^8$ s$^{-1}$ at standard conditions. Thus, $[Cu(NH_3)_2]^+$ complexes can be assumed to be paired and separated several times between each $O_2$ adsorption event. The short lifetime of the paired complexes put doubts on previous suggestions[16] that the SCR-activity is limited by the complex pairing.

To investigate the mobility of $[Cu(NH_3)_2]^+$ and $NH_4^+$, we analyze the mean square displacements (MSD), Fig. 6. Figure 6a, b compares the mobility of $[Cu(NH_3)_2]^+$ and $NH_4^+$ for Si/Al of 5, with three different Cu/Al ratios. For $[Cu(NH_3)_2]^+$, a high Cu/Al ratio yields a high mobility. The same trend is present for $NH_4^+$ ions, hence the presence of $[Cu(NH_3)_2]^+$ ions seem to enhance the diffusion. This could be attributed to the higher mobility of $[Cu(NH_3)_2]^+$ versus $NH_4^+$. As $[Cu(NH_3)_2]^+$ complexes has a higher mobility, $NH_4^+$ ions respond to the diffusion of $[Cu(NH_3)_2]^+$ by counter diffusion events. One example is illustrated in Fig. 3c, showing the case where the diffusion of one Cu-complex stimulates the diffusion of $NH_4^+$ to maintain local charge neutrality.

The result that a high Cu/Al ratio promotes Cu-ion mobility is consistent with operando electron paramagnetic resonance[16] and by impedance spectroscopy studies[23]. It should be noted that the diffusion of the charged species is slow with respect to neutral molecules[24]. In Fig. 6c, d, the MSD is evaluated for situations with the same number of $[Cu(NH_3)_2]^+$ ions but different Si/Al ratios. Interestingly, a lower Si/Al ratio leads to a lower diffusion for both $[Cu(NH_3)_2]^+$ and $NH_4^+$. The low diffusion is a consequence of $NH_4^+$, blocking the diffusion by occupying positions in the eight-membered rings, as evident from a pair correlation function (PCF) analysis (see Supplementary Fig. 9).

To summarize, the MSD analysis show that the amount of $[Cu(NH_3)_2]^+$ enhances the diffusion, whereas the presence of $NH_4^+$ inhibit both self- and $[Cu(NH_3)_2]^+$ diffusion. MSD analyses represent averages, thus, the individual mobility of each $[Cu(NH_3)_2]^+$ is not

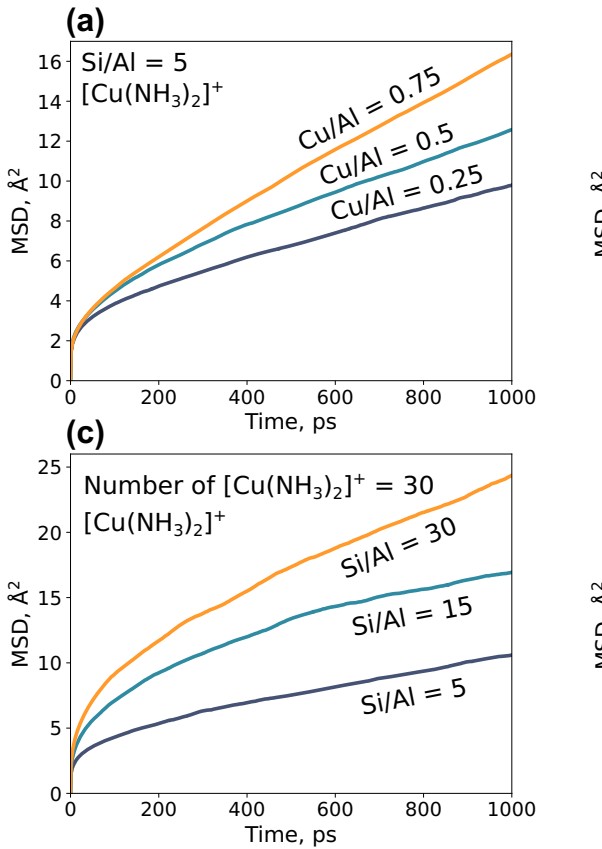

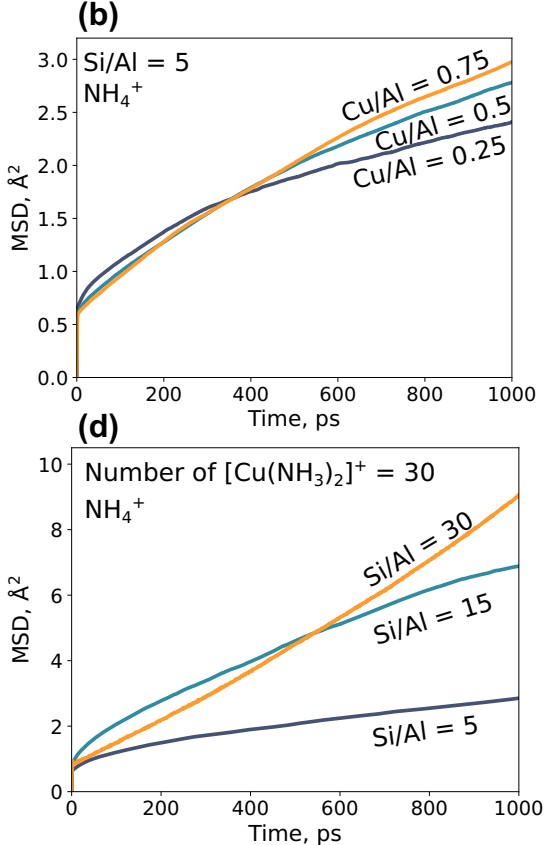

**Fig. 6 | Mean-square displacement analysis (MSD). a** $[Cu(NH_3)_2]^+$ and **b** $NH_4^+$ for Si/Al = 5 and Cu/Al = 0.25, 0.5, and 0.75. **c, d** are for the same Cu-loading, with Si/Al of 30, 15, and 5. Note that for (**d**), the number of $NH_4^+$ ions is low for the simulation with a high Si/Al ratio. The simulations have been performed with an H mass of 3 amu to enable large simulation times (5 ns). For a simple lattice gas, the MSD is proportional to $1/\sqrt{m}$,[45] where $m$ is the mass. Source data are provided as a Source Data file.

evident. The MSD evaluated for each Cu-complex reveals a large difference in the mobility of the complexes (see Supplementary Fig. 11). Some complexes stay in the initial cage during the 1 ns simulation, whereas other complexes move distances of 40 Å$^2$. The results demonstrate the heterogeneous nature of the Cu-CHA material, where only some Al-distributions promote the formation of $[Cu(NH_3)_2]^+$ pairs.

## Discussion

We have developed a machine-learning force field (ML-FF) augmented with long-ranged interactions to study the diffusion and pairing of $[Cu(NH_3)_2]^+$ in Cu-CHA. The ML-FF approach provides information that is inaccessible to experiments and DFT calculations. Using meta-dynamics simulations, we find that the barrier for $[Cu(NH_3)_2]^+$ depends sensitively on both the local and the distant Al-distributions. The presence of neighboring $[Cu(NH_3)_2]^+$ and $NH_4^+$ cations as well as the Al-distribution is found to significantly influence the dynamics of the $[Cu(NH_3)_2]^+$.

Using unbiased simulations, we find that a higher Cu-loading enhances the diffusion of both $[Cu(NH_3)_2]^+$ and $NH_4^+$ via counter diffusion mechanisms. We find that an increased Cu-loading also results in a larger fraction of paired $[Cu(NH_3)_2]^+$. A decreased Si/Al ratio is found to reduce the mobility of $[Cu(NH_3)_2]^+$ and $NH_4^+$, which is traced to hindered motions because of $NH_4^+$ species occupying positions in the eight-membered rings in the CHA. Simultaneously, increasing the Al-content enhances the pairing of $[Cu(NH_3)_2]^+$. A low Si/Al ratio can result in a higher fraction of paired $[Cu(NH_3)_2]^+$ complexes than estimated by a random distribution of $[Cu(NH_3)_2]^+$ complexes. The simulations reveal how the detailed Al-distribution results in a heterogeneous material where some distributions facilitate

$[Cu(NH_3)_2]^+$ pairing, whereas others do not. Our results provides unique information on the dynamics of $[Cu(NH_3)_2]^+$ complexes in Cu-CHA, and how the dynamics depend on materials properties such as Cu-loading, Si/Al ratio, and Al-distribution. Generally, the present study demonstrates the ability of machine-learning force field (ML-FF) augmented with long-ranged interactions to study dynamic phenomena in zeolites important for catalytic reactions. The advancement provides the possibility to study large systems over extended time scales at the accuracy of DFT calculations.

## Methods
### DFT calculations

The training data for the ML-FF is generated using density functional theory (DFT) calculations performed with the Vienna Ab initio Simulation Package (VASP)[25,26] version 5.4.4. The Perdew–Burke–Ernzerhof (PBE)[27] functional is used to approximate exchange-correlation effects. The PBE functional is augmented with Grimme's D3 approach[28] to account for dispersion forces and a Hubbard term (6 eV) on Cu to prevent over-delocalization of Cu 3d-states. PBE+U+D3 has previously been shown to describe the Cu-CHA system with reasonable accuracy[29]. The Kohn–Sham orbitals are expanded in plane waves with a cut-off value set to 480 eV. The interaction between the core and valence electrons is described using the projector augmented wave (PAW) method[30,31]. The PBE PAW (Cu, Si, Al, O, N, H) potentials version 54 is used. The considered valence electrons for the different elements are Cu(11), Si(4), Al(3), O(6), N(5), and H(1). The k-point sampling is restricted to the $\Gamma$ point. The convergence criteria for the SCF loop is set to $10^{-6}$ eV. The hexagonal CHA cage is used for training consisting of 36 Si and 72 O. One Si atom is replaced by an Al atom for each counter ion ($[Cu(NH_3)_2]^+$ and $NH_4^+$).

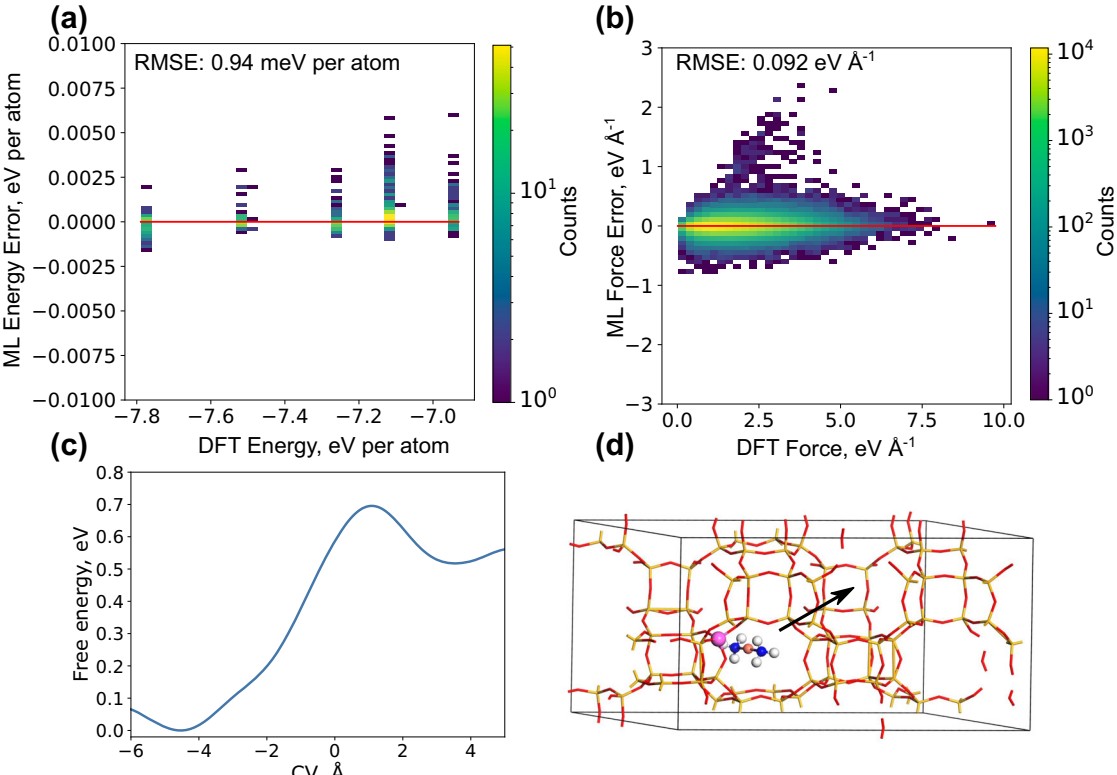

**Fig. 7 | Validation of the machine-learning force field (ML-FF). a** Correlation between predicted energies by ML-FF and density functional theory (DFT) results. **b** The root mean squared error (RMSE) is given for both energies and forces, calculated from $n = 700$ and $n = 23{,}850$ points, respectively. The ML error is the ML-FF predicted value subtracted from the DFT result. **c** Metadynamics simulation for the diffusion of $[Cu(NH_3)_2]^+$ in a $2 \times 1 \times 1$ unit cell as a function of the collective variable (CV). **d** Structure used for metadynamic simulation in (**c**), with an arrow indicating diffusion path. Atomic color codes: H (white), N (blue), O (red), Al (purple), Si (yellow), and Cu (bronze). Source data are provided as a Source Data file.

## Model training

The ML-FF is constructed using DeepMD-Kit[32] version 2.1.5, which uses a neural network. The cut-off radius is set to 6 Å with smoothing starting from 5 Å. The embedding net is set to three layers consisting of 32, 64, and 128 neurons, and the fitting net is set to three layers with 240 neurons each. The model is trained for $10^6$ steps, with the learning rate set to start at $5.0 \times 10^{-4}$ and finish at $5.0 \times 10^{-8}$ with exponential decay. Long-range interactions are augmented with a recently implemented approach[22] using charges distributed as spherical Gaussians on Cu, Al, N, and H. The hydrogen atom is divided into two atom types denoted $H_a$ and $H_b$, related to either $[Cu(NH_3)_2]^+$ or $NH_4^+$, respectively. The charges are evaluated using Bader-decomposition[33] version 1.04 and set to Al($-0.91$ e), Cu(0.53 e), N($-1.22$ e), $H_a$(0.47 e), and $H_b$(0.5325 e).

## Molecular dynamics

Molecular dynamics simulations using the ML-FF are performed with the large-scale atomic/molecular massively parallel simulator (LAMMPS)[34] 23 Jun 2022−Update 4. The temperature is set to 473 K in the NVT ensemble controlled by a Nosé−Hoover thermostat[35,36]. The mass of hydrogen is set to three amu to facilitate the integration of motion, allowing a time step of 1 fs. Well-tempered metadynamics[37] simulations are performed using the open-source, community-developed PLUMED library[38–40] version 2.8.2. The initial height and width of the Gaussian is set to 0.01 eV and 1.0 Å, respectively, and is deposited every 400th step with a bias factor of 10. For the multi-dimensional metadynamics simulation, the multiple walker[41] approach is used with three walkers. The collective variables used for $[Cu(NH_3)_2]^+$ and $NH_4^+$ are described in Supplementary Methods. The

minimum free energy paths, connecting the different minima are derived using the MEPSA[42] software version 1.4.

## Data collection

The training data is collected by constructing a preliminary ML-FF using structures extracted from an AIMD simulation. Subsequently, ML-FF simulations are performed, where structures are uniformly extracted along a trajectory, and single-point DFT calculations are performed. Metadynamics is employed to ensure adequate sampling of the diffusion of $[Cu(NH_3)_2]^+$ complexes through the eight-membered ring. Increasingly more complex structures are added by including more Al-sites and counter ions ($[Cu(NH_3)_2]^+$ and $NH_4^+$), and increasing the temperature from 25 °C to 300 °C. The approach resulted in a robust ML-FF. 90% of the data is used as training data and 10% as validation data. The final model includes training data consisting of about 52,000 single-point calculation from 160 different configurations that have a Si/Al ratio ranging from 4.4 to 107. See Supplementary Methods for further information on how the training data is collected.

## Validation

Having constructed an ML-FF, it is critical to examine the performance to validate that the force field reproduce DFT data not used in the training[43]. To do this, a range of molecular dynamic simulations is performed using the ML-FF, and configurations are extracted along the trajectory. The structures contain Al-distributions and situations on which the ML-FF has not been trained. This ensures that the ML-FF can extrapolate to Al distributions not present in the training data. For each structure, the energies and forces are evaluated using both DFT and ML-FF, see Fig. 7a, b. The calculated root mean squared value is

0.94 meV per atom for the energies and 0.092 eV Å$^{-1}$ for the forces, which is comparable or better compared to other studies.

To demonstrate that the ML-FF captures the long-range Coulomb interactions, single-point DFT calculations are performed for increasingly longer distances between Al and $[Cu(NH_3)_2]^+$ up to 20 Å. The ML-FF is capable of reproducing the long-ranged interaction. In addition, an ML-FF without explicit long-ranged interactions, is constructed for comparison. The exclusion of long-ranged interactions results in a lower energy penalty when a $[Cu(NH_3)_2]^+$ complex diffuses away from an Al ion as well as higher diffusivity for both $[Cu(NH_3)_2]^+$ and $NH_4^+$, see Supplementary Discussion.

Validation simulations are done also for free energy diffusion barriers. In this case, a $2 \times 1 \times 1$ unit cell is constructed, where $[Cu(NH_3)_2]^+$ is allowed to diffuse to an adjacent cell without any associated Al-ion, as illustrated in Fig. 7d with the free energy barrier in Fig. 7c. A negative CV corresponds to the $[Cu(NH_3)_2]^+$ complex being in the same cage as the Al-ion. This kind of simulations using AIMD with metadynamics or umbrella sampling has been performed previously[11,18] and serves as a test for the long-range interactions for the barriers as the distance between Al and Cu extends to 11 Å in the diffusion event, which is significantly longer than the 6 Å cut-off radius employed in the ML-FF. Our simulation estimates a free energy barrier of 0.70 eV, with the final state being 0.51 eV, less stable than the initial state. This is similar to previous simulations predicting the free energy barrier to be 0.57 or 0.79 eV[11,18].

## Reporting summary

Further information on research design is available in the Nature Portfolio Reporting Summary linked to this article.

## Data availability

The training data, validation data, and first and last structures from MD trajectories generated in this study have been deposited in the Zenodo database under accession code[44] https://doi.org/10.5281/zenodo.14290247. The source data are also provided in the Zenodo database.

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

## Acknowledgements

We acknowledge support from the European Union's Horizon 2020 research and innovation program under the Marie Sklodowska-Curie grant agreement no. 955839 (CHASS). Additional support from the Swedish Energy Agency (47110-1) is acknowledged. The calculations have been performed at C3SE (Göteborg), NSC (Linköping), and PDC (Stockholm) through a NAISS grant. The Competence Centre for Catalysis (KCK) is hosted by Chalmers University of Technology and financially supported by the Swedish Energy Agency (52689-1) and the member companies Johnson Matthey, Perstorp, Powercell, Preem, Scania CV, Umicore, and Volvo Group.

## Author contributions

All the authors conceived the study. J.D.B. developed the force field, performed the simulations, and drafted the manuscript. H.G. and M.V. supervised the work. H.G. edited the manuscript.

## Funding

## Competing interests

The authors declare no competing interests.
