## [Transparent Peer Review file · Nature Communications]

Influence of aluminium distribution on the diffusion mechanisms and pairing of $[\text{Cu}(\text{NH}_3)_2]^+$ complexes in Cu-CHA

Corresponding Author: Professor Henrik Gronbeck

Version 0:

Reviewer comments:

Reviewer #1

(Remarks to the Author)

The manuscript by Bjerregaard, Votsmeier, and Grönbeck introduces a machine learning force field (MLFF) augmented with explicit modeling of long-range electrostatic interactions to investigate the mobility and pairing of $[\text{Cu}(\text{NH}_3)_2]^+$ complexes in Cu-exchanged Chabazite (Cu-CHA). They report the free energy landscape of diffusion and pairing of $[\text{Cu}(\text{NH}_3)_2]^+$ complexes in Cu-CHA and demonstrate how these dynamics depend on material properties, including Cu-loading, Si/Al ratio, and Al-distribution. This work represents a significant advancement in understanding the performance of Cu-CHA for selective catalysis, and the proposal for explicit modeling of long-range interactions in the study of Cu-CHA is a noteworthy contribution to the field.

As I am not an expert on Cu-CHA, this review primarily focuses on the modeling and machine learning aspects of the work.

1. I concur with the authors that the explicit modeling of long-range interactions in MLFF is crucial for accurately calculating the free energy and diffusivity of charged complexes. However, the extent to which accuracy is improved by this consideration remains unclear. I am interested in whether the results would be quantitatively or qualitatively incorrect if a short-range MLFF model were used. A comparative study between short-range and long-range models would significantly strengthen the authors' argument. While I do not suggest reperforming all calculations with the short-range model, two representative comparisons—one on free energy and one on diffusivity—demonstrating the differences between the models would suffice.

2. First principles calculations:

- The k-space grid used in the calculations should be reported.
- Explicitly report the names of the VASP pseudopotentials (PPs) used. There are three oxygen PPs (excluding the GW versions) with six valence electrons (O, O_h, O_s). Which one was employed?
- Note that no VASP PP for nitrogen has six valence electrons.

3. The unit of the partial charges should be "e". Please add the unit, e.g., Al(-0.91) should be Al(-0.91e).

4. In the molecular dynamics simulations, the mass of hydrogen atoms was set to 3 amu. While this is acceptable for calculating equilibrium properties like free energy, the increased mass would slow down diffusion, thereby underestimating diffusivity. Please use 1 amu for the mean squared displacement (MSD) calculations and reduce the time-step to, for example, 0.5 fs to stabilize the integrator.

5. Please provide definitions for the collective variables used in the study.

6. The strategy for training data generation is not clear from the manuscript:

- Were metadynamics MLFF simulations used to explore the configurations?
- How were the configurations selected from the trajectories? Were they chosen uniformly or using uncertainty estimation techniques like query-by-committee?

Overall, this manuscript presents a valuable contribution to the field, and addressing the points above would further enhance its clarity and impact.

Reviewer #2

(Remarks to the Author)

The study entitled "Collective effects and aluminum distribution control diffusion and pairing of $[\text{Cu}(\text{NH}_3)_2]^+$ complexes in Cu-CHA" focuses on Cu-exchanged Chabazite (Cu-CHA) as a catalyst for selective catalytic reduction (SCR) of NO_x using ammonia. In particular, the authors develop a machine learning potential to investigate the diffusion of $[\text{Cu}(\text{NH}_3)_2]^+$ complexes and their pairing, which is influenced by the local environments within the zeolite framework and the consequent implications for catalytic activity. The research employs computational methods such as DFT, molecular dynamics and machine learning. The work is original and the manuscript is well written which include results using state-of-the-art methodologies for an industrially relevant catalytic reaction.

The study reveals that the pairing of $[\text{Cu}(\text{NH}_3)_2]^+$ complexes is affected by the local distribution of Al-ions, with higher Cu/Al ratios leading to increased pairing fractions. It demonstrates that the stability of paired complexes is influenced by the Si/Al ratio, with lower ratios promoting pairing due to electrostatic stabilization of the paired complexes. The lifetime of paired complexes varies significantly, indicating that local environments play a crucial role in their stability and the overall SCR reaction kinetics.

There are, however, some aspects that need clarification before considered for publication:

1) In the support information the authors explain the collective variable (CV) used in metadynamics simulations. It is explicitly stated the CV used for the diffusion of $[\text{Cu}(\text{NH}_3)_2]^+$ as follows: "The CV describes the diffusion of the $[\text{Cu}(\text{NH}_3)_2]^+$ complex through the eight-membered ring connecting the two cages ...". However, in Figure 4 a two-dimensional plot is shown including a CV for the mobility of NH_4^+ as well. Which CV was used for NH_4^+ ? Was it the same one used for $[\text{Cu}(\text{NH}_3)_2]^+$? This should be explicitly mentioned.

2) It is not mentioned how was the free energy of activation calculated. An approach quite frequent in the literature is to take the difference between the maximum and minimum point of the free energy surface (FES). Although not critical this approach could have some error involved. Was the activation free energy calculate as a difference between maximum and minimum points of the FES? Or the minimum basin was integrated to derive the actual number?

3) The authors make a valid point in the introduction mentioning that the ML force fields suffer from a lack of proper long-range interaction description since those are based on a local atomic description. They also state that the assumption of locality is not applicable to Cu-CHA. Consequently they use a point charge model to better describe electrostatic interactions as stated in the last paragraph of the introduction. I have to comments in this respect. One the one hand, point charges are the simplest approximation (generally used in classical force fields) for the electrostatic interactions. One the other hand, it is not explicitly addressed in the manuscript how point charges improve the results. It would be relevant to have address this, since it appears to be part of the methodological novelty of the study. Can the authors elaborate (if possible comparing the literature or quantified with and without the this correction) how this explicit description of electrostatic interactions impact the results? For example, how different the activation free energies would be, how it would affect the stability of paired complexes. Is this description necessary/crucial for this particular system?

4) Finally, as minor detail, it might be convenient to update the label of the x-axis in Figure 4b to explicitly show that it refers to the CV describing the $[\text{Cu}(\text{NH}_3)_2]^+$ diffusion.

Version 1:

Reviewer comments:

Reviewer #1

(Remarks to the Author)

The revised manuscript has successfully addressed all of my concerns and is now recommended for acceptance without further changes.

Reviewer #2

(Remarks to the Author)

I thank the authors for addressing the concerns of the reviewer. I recommend the paper for publication provided the authors include the error in the newly added Supplementary Fig. 7. I suggest to include the sampling error of the metadynamics simulations to properly assess what part of the difference between the two plots corresponds to the inclusion of the long-range interactions explicitly and what has to be attributed to just the error of the sampling method.

Reviewer #1:

The manuscript by Bjerregaard, Votsmeier, and Grönbeck introduces a machine learning force field (MLFF) augmented with explicit modeling of long-range electrostatic interactions to investigate the mobility and pairing of $[\text{Cu}(\text{NH}_3)_2]^+$ complexes in Cu-exchanged Chabazite (Cu-CHA). They report the free energy landscape of diffusion and pairing of $[\text{Cu}(\text{NH}_3)_2]^+$ complexes in Cu-CHA and demonstrate how these dynamics depend on material properties, including Cu-loading, Si/Al ratio, and Al-distribution. This work represents a significant advancement in understanding the performance of Cu-CHA for selective catalysis, and the proposal for explicit modeling of long-range interactions in the study of Cu-CHA is a noteworthy contribution to the field.

As I am not an expert on Cu-CHA, this review primarily focuses on the modeling and machine learning aspects of the work.

1. I concur with the authors that the explicit modeling of long-range interactions in MLFF is crucial for accurately calculating the free energy and diffusivity of charged complexes. However, the extent to which accuracy is improved by this consideration remains unclear. I am interested in whether the results would be quantitatively or qualitatively incorrect if a short-range MLFF model were used. A comparative study between short-range and long-range models would significantly strengthen the authors' argument. While I do not suggest reperforming all calculations with the short-range model, two representative comparisons—one on free energy and one on diffusivity—demonstrating the differences between the models would suffice.

We thank the Reviewer for highlighting this important point and agree that we originally did not present clear evidence for the importance of long-ranged interactions. For the revised manuscript, we have trained a new ML-FF using the same training data and force-field settings as the original ML-FF excluding explicit long-ranged interactions. The ML-FFs with and without long-ranged interactions is here referred to as long ranged deep potential (DPLR) force-field and deep potential (DP) force-field. We compare the performance of the force-fields in both free energy calculations and unbiased MD simulations.

In the first comparison, we consider meta-dynamics simulations of $[\text{Cu}(\text{NH}_3)_2]^+$ complex pairing (Case VI Figure 2a of the manuscript). The comparison is shown in Figure R1a. The DP model predicts a relative free energy increase of 0.41 eV compared to 0.68 eV predicted by the DPLR model. This is a large difference, which corresponds to a difference in 10^3 in population of the paired configuration using a Boltzmann distribution at 473 K.

To further investigate the long-ranged interactions, we conducted free energy calculation for a system with a single Al ion, allowing the $[\text{Cu}(\text{NH}_3)_2]^+$ complex to diffuse two cages away from its starting cage with Al, Figure R1b. The trend remains consistent with the results in Figure R1a, with the DPLR model predicting a higher energy. In the second cage, the DP model predicts a relative free energy of 0.52 eV, while the DPLR model predicts 0.62 eV.

Figure R1. Comparison DPLR and DP.

The importance of including the long-ranged interactions is exemplified also in simulations of the mean-square displacements, Figure R2. The DP force-field shows a higher diffusivity than the DPLR force-field. This is in line with the free energy landscapes in Figure R1, where the DP model predicts a lower free energy change for the diffusion.

Figure R2. Comparison DPLR and DP.

A detailed discussion on the importance of including long-ranged interactions has been added to the revised SI. Moreover, the following comment has been added on page 12 (*Methods – Validation*) in the manuscript.

“In addition, an ML-FF without explicit long-ranged interactions, is constructed for comparison. The exclusion of long-ranged interactions results in a lower energy penalty when a $[\text{Cu}(\text{NH}_3)_2]^+$ complex diffuses away from an Al ion as well as higher diffusivity for both a $[\text{Cu}(\text{NH}_3)_2]^+$ and NH_4^+ , see Supplementary Discussion.”

2. First principles calculations:

- The k-space grid used in the calculations should be reported.
- Explicitly report the names of the VASP pseudopotentials (PPs) used. There are three oxygen PPs (excluding the GW versions) with six valence electrons (O, O_h, O_s). Which one was employed?
- Note that no VASP PP for nitrogen has six valence electrons.

We agree that too many details in the DFT methodology were omitted. In the revised manuscript, k-space grid has been added together with the used pseudopotentials. We have also corrected the mistake in the number of valence electrons for nitrogen. The following changes have been done on page 10 in section *Methods - DFT calculations*:

“The PBE PAW (Cu, Si, Al, O, N, H) potentials version 54 is used.”

“The k-point sampling is restricted to the Γ -point.”

“N(5)”

3. The unit of the partial charges should be "e". Please add the unit, e.g., Al(-0.91) should be Al(-0.91e).

We have added the unit for the partial charges on page 10 in section *Methods – Model training*

4. In the molecular dynamics simulations, the mass of hydrogen atoms was set to 3 amu. While this is acceptable for calculating equilibrium properties like free energy, the increased mass would slow down diffusion, thereby underestimating diffusivity. Please use 1 amu for the mean squared displacement (MSD) calculations and reduce the time-step to, for example, 0.5 fs to stabilize the integrator.

We thank the Reviewer for bringing up this point. The higher mass for H was used to be able to use a larger time step in the simulations, thus, being able to run long trajectories. Each case in Figure 6 is based on simulations for 5 ns, which each require about 10 days using 128 cores. We have in the revised manuscript performed additional simulations using the mass of hydrogen set to 1.00784 amu and a time step of 0.5 fs. This simulation was done for the case of Si/Al = 5 and Cu/Al = 0.5. The results are shown in Figure R3.

Figure R3. Effect on H-mass in the MSD simulations.

The diffusion of the $[\text{Cu}(\text{NH}_3)_2]^+$ complex is not significantly influenced by the choice of mass, which is a consequence of the high mass of Cu. As expected, the results for NH_4^+ is affected by the change in hydrogen mass. A lower mass speeds up the diffusion. The mean square displacement can be related to the diffusion coefficient (D) according to:

$$MSD = 2Dt$$

The diffusion coefficient scales in a lattice gas model by $1/\sqrt{m}$. Thus, the ratio of the diffusion coefficients is simplistically:

$$\frac{D_i}{D_j} = \left(\frac{m_j}{m_i}\right)^{1/2}$$

Clearly, the square-root dependence could be modified for more complicated situations as in the present paper. Based on this relationship, our mean squared displacement analysis for NH_4^+ , while being underestimated, is expected to capture the correct trends, for different Cu/Al and Si/Al ratios.

We have in the revised manuscript added a comment in the caption of Figure 6 that the MSD are underestimated for NH_4^+ diffusion due to the higher H-mass (page 21):

“The long simulation times (5ns) are possible by using an H mass of 3 amu. For a simple lattice gas the MSD is proportional to $1/\sqrt{m}$.”

5. Please provide definitions for the collective variables used in the study.

We agree that the collective variables were not completely described and have extended the section *Collective variables* in the SI, along with the following comment on page 11 in section *Methods – Molecular dynamics*

“The collective variables used for $[\text{Cu}(\text{NH}_3)_2]^+$ and NH_4^+ are described in the Supplementary Methods.”

6. The strategy for training data generation is not clear from the manuscript:

- Were metadynamics MLFF simulations used to explore the configurations?
- How were the configurations selected from the trajectories? Were they chosen uniformly or using uncertainty estimation techniques like query-by-committee?

Metadynamics is used to sample rare events involving the diffusion of $[\text{Cu}(\text{NH}_3)_2]^+$ through the eight-membered. This ensures that the ML-FF has been properly trained for this critical events.

The training data has been selected uniformly from molecular dynamics trajectories and the procedure for collecting training data can be found in SI. To clarify the data collection procedure, the following are added on page 11, in the section *Methods - Data collection*.

“Subsequently, ML-FF simulations are performed, where structures are uniformly extracted along the trajectory, and single-point DFT calculations are performed. Metadynamics is

employed to ensure adequate sampling of the diffusion of $[\text{Cu}(\text{NH}_3)_2]^+$ complexes through the eight-membered ring”

In the end of the *Methods - Data collection* section, a reference is made to the Supplementary Methods.

“For further information on how the training data is collected, see Supplementary Methods.”

Overall, this manuscript presents a valuable contribution to the field, and addressing the points above would further enhance its clarity and impact.

Reviewer #2:

The study entitled “Collective effects and aluminum distribution control diffusion and pairing of $[\text{Cu}(\text{NH}_3)_2]^+$ complexes in Cu-CHA” focuses on Cu-exchanged Chabazite (Cu-CHA) as a catalyst for selective catalytic reduction (SCR) of NO_x using ammonia. In particular, the authors develop a machine learning potential to investigate the diffusion of $[\text{Cu}(\text{NH}_3)_2]^+$ complexes and their pairing, which is influenced by the local environments within the zeolite framework and the consequent implications for catalytic activity. The research employs computational methods such as DFT, molecular dynamics and machine learning. The work is original and the manuscript is well written which include results using state-of-the-art methodologies for an industrially relevant catalytic reaction.

The study reveals that the pairing of $[\text{Cu}(\text{NH}_3)_2]^+$ complexes is affected by the local distribution of Al-ions, with higher Cu/Al ratios leading to increased pairing fractions. It demonstrates that the stability of paired complexes is influenced by the Si/Al ratio, with lower ratios promoting pairing due to electrostatic stabilization of the paired complexes. The lifetime of paired complexes varies significantly, indicating that local environments play a crucial role in their stability and the overall SCR reaction kinetics.

There are, however, some aspects that need clarification before considered for publication:

1) In the support information the authors explain the collective variable (CV) used in metadynamics simulations. It is explicitly stated the CV used for the diffusion of $[\text{Cu}(\text{NH}_3)_2]^+$ as follows: “The CV describes the diffusion of the $[\text{Cu}(\text{NH}_3)_2]^+$ complex through the eight-membered ring connecting the two cages ...”. However, in Figure 4 a two-dimensional plot is shown including a CV for the mobility of NH_4^+ as well. Which CV was used for NH_4^+ ? Was it the same one used for $[\text{Cu}(\text{NH}_3)_2]^+$? This should be explicitly mentioned.

We thank the Reviewer for pointing out that we missed reporting also the collective variable for NH_4^+ . In the revised SI, have extended the section *Collective variables* to also include the CV for the NH_4^+ ion and added a sentence in section *Methods – Molecular dynamics* on page 11

“The collective variables used for $[\text{Cu}(\text{NH}_3)_2]^+$ and NH_4^+ are described in Supplementary Methods.”

2) It is not mentioned how was the free energy of activation calculated. An approach quite frequent in the literature is to take the difference between the maximum and minimum point of the free energy surface (FES). Although not critical this approach could have some error involved. Was the activation free energy calculate as a difference between maximum and minimum points of the FES? Or the minimum basin was integrated to derive the actual number?

The free energy barriers are calculated by taking the difference between the maximum and minimum point on the potential energy surface along the diffusion path. The following sentence is added on page 5 to clarify this.

“The free energy barrier is computed by taking the difference between the maximum and minimum points on the potential energy surface along the diffusion path.”

3) The authors make a valid point in the introduction mentioning that the ML force fields suffer from a lack of proper long-range interaction description since those are based on a local atomic description. They also state that the assumption of locality is not applicable to Cu-CHA. Consequently they use a point charge model to better describe electrostatic interactions as stated in the last paragraph of the introduction. I have to comments in this respect. One the one hand, point charges are the simplest approximation (generally used in classical force fields) for the electrostatic interactions.

We thank the Reviewer for the comment. In the original manuscript we described the charge correction as point charges. However, the corrections are spherical Gaussians with a certain spread parameter. This has been clarified in the revised manuscript page 10.

“Long-range interactions are augmented with a newly implemented approach²² using charges distributed as spherical Gaussians on Cu, Al, N and H.”

One the other hand, it is not explicitly addressed in the manuscript how point charges improve the results. It would be relevant to have address this, since it appears to be part of the methodological novelty of the study. Can the authors elaborate (if possible comparing the literature or quantified with and without the this correction) how this explicit description of electrostatic interactions impact the results? For example, how different the activation free energies would be, how it would affect the stability of paired complexes. Is this description necessary/crucial for this particular system?

We thank the Reviewer for addressing this point, which triggered us to develop a new ML-FF without the long-ranged correction. This point is the same as point 1 by Reviewer 1 and we refer to that response.

4) Finally, as minor detail, it might be convenient to update the label of the x-axis in Figure 4b to explicitly show that it refers to the CV describing the $[\text{Cu}(\text{NH}_3)_2]^+$ diffusion.

It is true that the reaction coordinate in Figure 4b mainly describes the diffusion of the $[\text{Cu}(\text{NH}_3)_2]^+$ complex. However, the reaction coordinate corresponds to the computed minimum free energy path between two minima on a three-dimensional free energy surface and could for other system involve the simultaneous movement of the two collective variables. To clarify this, we have changed the label for the x-axis in Figure 4b to “Minimum free energy path”

Reviewer #2 (Remarks to the Author):

I thank the authors for addressing the concerns of the reviewer. I recommend the paper for publication provided the authors include the error in the newly added Supplementary Fig. 7. I suggest to include the sampling error of the metadynamics simulations to properly assess what part of the difference between the two plots corresponds to the inclusion of the long-range interactions explicitly and what has to be attributed to just the error of the sampling method.

Response

An assessment of the error in the metadynamics sampling is not straight forward as the true potential energy surface is unknown. However, the error can be estimated from convergence of the relative free energy differences during the metadynamics trajectories. The Figure below shows (a) the free energy potential energy curve for one case of Cu-complex diffusion (Si/Al of 13). (b) shows the energy difference between position I and III, whereas (c) shows the energy difference between position I and II. The analysis shows that the energy difference between the two minima (b) is converged within about 0.1 eV for both potentials and that the difference between the potentials are larger than the error. Thus, the energy difference between the two potentials is due to the inclusion of the long-range interactions. The energy difference between positions I and II (c) converges faster and the error is smaller than in (b).

We have in the Supplementary Information added a comment on page 11 on the estimated error in the reported free energy curves:

“The errors in the free energy landscapes were estimated by analyzing the evolution of the free energy differences along the trajectories. The errors were found to be within about 0.1 eV.”